# Prevalence of oncogenic human papillomavirus (HPV 16/18) infection, cervical lesions and its associated factors among women aged 21–49 years in Amhara region, Northern Ethiopia

**Minwuyelet Maru Temesgen**[1]*, **Tefera Alemu**[1], **Birtukan Shiferaw**[1], **Seid Legesse**[1], **Taye Zeru**[2], **Mahteme Haile**[2], **Tesfaye Gelanew**[3]

1 Amhara Public Health Institute Dessie Branch, Dessie, Ethiopia, 2 Amhara Public Health Institute, Bahir Dar, Ethiopia, 3 Armauer Hansen Research Institute, Addis Ababa, Ethiopia

* minwuyelet@yahoo.com

## Abstract

### Background

Human papillomavirus (HPV) infection is considered as the major risk factor for the development of cervical cancer, second most frequent cancer in Ethiopia. However, the magnitude of the problem and the associated factors remain unrevealed in the Amhara region. This study aimed to determine the prevalence of HPV infection and factors contributing to the progression of HPV infection to cervical cancer.

### Methods

Facility-based cross-sectional study design was employed among women aged 21 to 49 years of age who came for routine cervical cancer screening to 4 randomly selected hospitals (2 general and 2 referral) of Amhara region from May to October, 2019. The sample size was calculated by using the single population proportion formula, proportionated to hospitals, and women were recruited consecutively. Socio demographic and clinical data were collected using a pretested questionnaire and detection of HPV infection was done using HPV test (OncoE6™ Cervical Test) specific to HPV16/18 in cervical swabs. Visual inspection with acetic acid (VIA) was used to determine cervical lesions (precancerous and cancerous). Descriptive statistics and bivariate and multivariate logistic regression were used to describe HR-HPV and cervical lesions burden and association between HR-HPV, and cervical lesions and potential risk factors.

### Results

Among 337 women 21 to 49 years (median age of 35 years ±SD = 7.1 years) of age enrolled in the study, The overall prevalence of oncogenic HPVs (HPV16/18) and the VIA-positivity rate, possible an indicative of cervical lesions, were 7.1% and 13.1%, respectively. Logistic

**Data Availability Statement:** All relevant data are within the manuscript and its Supporting Information files.

**Funding:** The authors received no specific funding for this work.

**Competing interests:** The authors have declared that no competing interests exist.

**Abbreviations:** APHI, Amhara Public Health Institute; HIV, Human Immunodeficiency Virus; HPV, Human Papillomavirus; HR, High risk; LR, Low risk; STI, Sexually transmitted illness; VIA, Visual Inspection with Acetic acid; WHO, World Health Organization.

regression analysis showed a significant association between early age of first sexual intercourse (COR = 2.26; 95% CI: 1.0–5.05) and level of education (COR = 0.31; 95% CI: 0.12–0.78) with cervical lesions. Higher odds of HPV positivity (COR = 1.56; 95% CI: 0.59–4.11, p = 0.36) and VIA positivity (COR = 1.39; 95% CI: 0.64–3.00, p = 0.39) were observed among participants who had a history of sexually transmitted illnesses (STIs).

## Conclusions

There was a relatively low prevalence of oncogenic HPV 16/18 and VIA-positivity in women attending four hospitals in the Amhara Region. Early age sexual contact, high parity, and being uneducated/low level of education were independently associated factors with HR-HPV infection and development of cervical lesions, highlighting the importance of prioritizing the limited HPV testing to those risk groups.

## Introduction

Human papillomavirus (HPV) comprises over 150 related DNA viruses that spreads sexually and through skin to skin contacts. Based on their virulence, these viruses are generally classified as high risk and low risk (LR) types [1]. Often our immunity can spontaneously clear off most of the HPV infections without treatment. However, infections with high risk HR-HPV types, especially in immune-compromised, can persists and possibility leads to warts, precancerous lesions or cancers [2]. That is why chronic infections with HR-HPV types are deep-rooted in almost all cervical cancer cases.

In the last six decades, the world has made significant progress towards the reduction of morbidity and mortality associated with cervical cancer. Despite this, cervical cancer remains a major threat to women's health, especially in resource-limited countries due to poor awareness and late detection of cervical lesions. For instance, 85% of deaths from cervical cancer occurred in low and middle-income countries [3]. Besides, 14% of global incidences and 18% of deaths occurs in Sub-Saharan African countries [4].

In Ethiopia cervical cancer is a major cause of morbidity and mortality of women of age 15–49 years with prevalence ranging from 14% to 17% [5–7]. Several factors have been identified to be risk factors for developing cervical cancer associated with HPV infections. These factors can be but not limited to poor awareness, socioeconomic and, cultural factors, lack of primary and secondary prevention methods, poor nutrition and other sexual transmitted infections (STIs: HIV infection, trichomoniasis, syphilis, hepatitis B and hepatitis C).

The World Health Organization (WHO) 2013 cervical cancer guideline recommended a regular screening for woman of reproductive age using visual inspection with acetic acid (VIA) or when possible HPV testing followed by cryotherapy treatment, in resource- limited setting. VIA alone is widely used for presumptive diagnosis of cervical cancer in extreme resource-limited settings, where cervical cancer screening by cervical cytology and HPV testing is unavailable [8]. The Ethiopia cervical cancer prevention and control 2015 guideline indicated three components of cervical cancer prevention and control. The first is primary prevention which focuses on decreasing HPV infection through behavioral changes or vaccination. The secondary prevention targets on screening and treating precancerous lesions and tertiary care is the third which contains management of invasive cervical cancer [9]. Several studies have shown the likelihood of unvaccinated HPV-negative women to develop cervical

cancer in the next 5–10 years is less, suggesting primary HPV testing is an essential preventive approach, particularly among unvaccinated women [10]. In Ethiopia, vaccination of girls prior to sexual debut using bivalent HPV vaccine that protects against HPV genotypes 16 and 18 was started among school girls as a pilot including Amhara region in 2019. However, the vaccines target certain genotypes and leave some proportion of the population infected by other genotypes unprotected. In addition, the vaccination coverage is very low and implementing this intervention is challenging in the regions where access to HPV-based testing is limited. Although, screening and treating cervical lesions using VIA remain crucial for successful cervical cancer prevention and control programs, it has some drawbacks related to sensitivity and specificity. Studies have shown VIA has about 67 to 79% sensitivity and 83 to 84% specificity [11, 12]. Therefore, screening with VIA together with HPV testing would be the best method for better outcome. Nonetheless, the proportion of HPV infection and cervical lesions (precancerous and cancerous lesions) among reproductive-age women as well as the associated factors remain unknown in the Amhara region despite this information is broadly seen as indispensable to the cervical cancer prevention programs. The main aim of this study was therefore to assess the magnitude of VIA positivity rate and prevalence of HR-HPV infection. The secondary aim was to identify associated potential risk factors to the development of precancerous or cancerous lesions.

## Materials and methods

### Study design and setting

A facility based cross-sectional study was conducted from May to October 2019 in four randomly selected government hospitals in the Amhara region. These hospitals were Tefera Hailu Memorial General Hospital, Felegehiwot Referral Hospital, Debretabor General Hospital and Debremarkos Referral Hospital that provides both primary HPV testing and cervical cancer screening and family planning services. Amhara Region is the second-largest populous region in Ethiopia which has 12 zones, 3 city administrations, and 180 districts (139 rural and 41 urban) and has 80 hospitals (5 referral, 2 general, and 73 primaries), 847 health centers, and 3,342 health posts. Screening women of reproductive age for cervical lesions using VIA in the region has been scaled up to all hospitals since 2015. However, HPV testing has been done in general and referral hospitals.

### Study population

All women who came for routine gynecologic or family planning services to those hospitals were used as a source population for this study. Women of age 21 to 49 years who were referred to the routine cervical cancer screening services in the study period were invited to participate and consented after being screened by VIA and HPV testing. However, women with confirmed cancerous lesions and pregnancy were excluded from the present study to avoid unnecessary discomforts related to the procedure's endocervical swabs.

### Sample size and sampling technique

Sample size was determined using a single population proportion formula by considering 5% margin of error, 95% confidence level, 16% prevalence of HPV infection in Ethiopia, 5% non-response rate and 1.5 design effect and women were recruited consecutively [13]. The total sample size, 337, was proportionated to the selected participating hospitals based on their clients' size: Tefera Hailu Memorial General Hospital (n = 51), Felegehiwot Referral Hospital

(n = 118), Debretabor General Hospital (n = 65), and Debremarkos Referral Hospitals (n = 103).

**Data collection.** Ten data collectors (1 nurse and 1 laboratory technician) from each study site and four supervisors from Amhara Public Health Institute (APHI) were recruited. Scio-demographic and clinical data were collected by study nurses using a structured and pre-tested questionnaire through face to face interviewing each one of the participant women. The pre-test was conducted on 5% of the study population. All clients of reproductive age who were tested by routine VIA were invited to participate in this study, and were screened for HPV infection after being consented and interviewed.

**Diagnosis of cervical lesions using visual inspection with acetic acid.** After cleaning away any extra mucus with a cotton swab, a five percent acetic acid solution was applied to the cervix for VIA. After 30 to 60 seconds, abnormal cervical tissues (e.g., pre-cancerous or cancerous lesions) turned white [14]. On the other hand, no aceto white lesions was observed on the VIA of normal cervical tissue.

**E6 HPV 16/18 oncoprotein testing.** Endocervical swab was collected from study participant women by inserting swab devices into endocervix and rotating them counter-clockwise to make full circle three times. Detection of HPV types 16 and 18 in the cervical swabs was performed using E6 HPV 16/18 oncoprotein detection lateral flow (LF) strip test (OncoE6[TM] Cervical Test (Arbor Vita Corporation, Fremont, CA, USA) as per manufacturer's instruction. Interpretation of the test result was also done according to the manufacturer's instruction.

**Data analysis.** Collected data were checked for completeness before data entry. Data were cleaned, coded and entered in to Epidata 3.1 software and exported to STATA version 14 for analysis. Frequencies, proportion and summary statistics were used to describe the study population with relevant variables. Logistic regression was used to identify factors associated with HPV infection and odds ratio with 95% CI was used to assess the degree of association. P value $<0.05$ is considered statistically significant association and variables with $P<0.2$ were tested for multivariable logistic regression.

**Ethical considerations.** The study was approved by the Amhara Public Health Institute Ethical Review Board (Approval Number: 03/379/2011). Written permissions were obtained from zonal and district health authorities and the respective hospitals. All study participants were informed about the purpose of the study, and written consent (finger print for those who cannot read and write) was taken from each one of them. Confidentiality of any information related to the participants was maintained by excluding personal identifiers. Participants with a positive HPV test were linked to clinicians for further treatment and follow-up.

## Results

### Socio-demographic and clinical characteristics of participant women

A total of 337 women 21 to 49 years of age were enrolled in the study with median age of 35 years (±SD = 7.1 years). Of which 189 (56.1%) were rural residents and 182 (54.0%) participants were below primary education. Regarding occupational status 130 (38.6%) were house wives. The majority, 257 (76.3%) of participants were married whereas 45 (13.3%) were divorced (Table 1).

The mean number of children was 3.8 (±SD = 2.3) ranging from 1 to 9 with mean family size of 4.3 (±SD = 2.1) ranging from 1 to 13. All study participant women (n = 337) were screened for HIV and assessed for history of STIs when enrolled in this study and 67 (19.9%) and 61 (18.1%) of them had HIV and history of STIs, respectively. Two hundred fifty-seven (76.3%) of participants had a history of contraceptive use and of which 16.8% reported oral contraceptive use for 1 to 25 years with mean of 3.2 years (±SD = 3.9). Majority, 255 (75.7%) of

study participants had heard about cervical cancer while 161 (47.8%) had not heard about symptoms of cervical cancer. Two hundred ninety-nine (88.7%) never heard about HPV and only 10 (2.9%) had got vaccinated for HPV. Most importantly, only 52 (15.4%) of study participants reported being screened for HPV infection (Table 2).

## HR-HPV infection, cervical lesions and associated factors

All the 337 participant women had successful HPV and VIA tests. Of which 24 (7.12%) and 44 (13.05%) women had positivity to HR-HPV and VIA tests, respectively. As shown in Tables 3 and 4, the proportion of HPV infection and VIA-positivity was shown to be evenly distributed across different age groups, however, relatively higher HPV and VIA positivity rates were observed among older participants (41–49 years).

The prevalence of HR-HPV 16/18 infections among women with and without cervical lesions was 38.6% and 2.4%, respectively (Table 4). Moreover, VIA identified 17 and missed 7 out of 24 HPV positive women that make its sensitivity 70.8%. Although its positive predictive value is 61.4% (27 out of 44), 286 out of 293 participant women were correctly identified negative which make VIA sensitivity 91.3% The prevalence of HR-HPV 16/18 infections in HIV-negative and HIV-positive women was 8.1% and 3.0%, respectively (Table 5).

Multivariate logistic regression analysis revealed slightly higher odds of HPV infection in study participants who had history of STIs, and who started first sexual contact when they were under 18 years of age as compared to their counter parts (Table 5).

Significant association between cervical lesions with educational status and age of first sexual intercourse was observed using multivariate logistic regression analysis. Participant

**Table 1. Demographic characteristics of participant women (n = 337) aged 21–49 years, Amhara region, Ethiopia, 2019.**

| Characteristic | Frequency (N = 337) | Percent |
|---|---|---|
| Age in years | | |
| 21–30 | 112 | 33.2 |
| 31–40 | 158 | 46.9 |
| 41–49 | 67 | 19.9 |
| Education | | |
| Primary and below | 182 | 54.0 |
| Secondary education | 53 | 15.7 |
| College and above | 102 | 30.3 |
| Occupation | | |
| House wife | 130 | 38.6 |
| Employee | 102 | 30.3 |
| Farmer | 54 | 16.0 |
| Merchant | 31 | 9.2 |
| Other | 20 | 5.9 |
| Marital status | | |
| Never married | 27 | 8.0 |
| Married | 257 | 76.3 |
| Divorced | 45 | 13.3 |
| Widowed | 8 | 2.4 |
| Residence | | |
| Urban | 148 | 43.9 |
| Rural | 189 | 56.1 |

**Table 2. Awareness of cervical cancer among participant women (n = 337) aged 21–49 years, Amhara region, Ethiopia, 2019.**

| Characteristic | Frequency (N = 337) | Percent |
|---|---|---|
| Heard about cervical cancer | | |
| Yes | 255 | 75.7 |
| No | 82 | 24.3 |
| Heard about symptoms of cervical cancer | | |
| Yes | 176 | 52.2 |
| No | 161 | 47.8 |
| Heard about HPV | | |
| Yes | 299 | 88.7 |
| No | 38 | 11.3 |
| Vaccinated for HPV | | |
| Yes | 10 | 2.9 |
| No | 327 | 97.1 |
| Screened for HPV infection | | |
| Yes | 52 | 15.4 |
| No | 285 | 84.6 |

women with college degree or higher education had 0.31 reduced odds of cervical lesions than women who had primary education or less. Women who had started sexual contact at age of less than 18 years old had 2.26 times more likely to have cervical lesions than their counter parts (Table 6). The level of education was also found to be significantly associated with knowledge of cervical cancer indicated by chi square (p<0.001). One hundred fourteen (62.6%)

**Table 3. Proportion of HPV infection and cervical lesions among participant women (n = 337) aged 21–49 years, Amhara region, Ethiopia, 2019.**

| Characteristic | Number (N = 337) | HPV infection based on E6 16/18 antigen test | | VIA cervical cancer screening | |
|---|---|---|---|---|---|
| | | Positive | Negative | Positive | Negative |
| | | N (%) | N (%) | N (%) | N (%) |
| **Age in years** | | | | | |
| 21–30 | 112 | 8 (7.1) | 104 (92.9) | 9 (8.1) | 103 (91.9) |
| 31–40 | 158 | 9 (5.7) | 149 (94.3) | 18 (11.4) | 140 (88.6) |
| 41–49 | 67 | 7 (**10.4**) | 60 (89.6) | 17 (**25.4**) | 50 (74.6) |
| Overall | 337 | 24 (**7.1**) | 313 (92.9) | 44 (**13.1**) | 293 (86.9) |
| **Education** | | | | | |
| Primary and below | 182 | 15 (8.2) | 167 (91.8) | 30 (16.5) | 152 (83.5) |
| Secondary education | 53 | 7 (**13.2**) | 46 (86.8) | 8 (15.1) | 45 (84.9) |
| College and above | 102 | 2 (1.9) | 100 (98.1) | 6 (5.9) | 96 (94.1) |
| Overall | 337 | 24(**7.1**) | 313(92.9) | 44(**13.1**) | 293(86.9) |
| **Marital status** | | | | | |
| Never married | 27 | 5 (**18.5**) | 22 (81.5) | 4 (**14.8**) | 23 (85.2) |
| Married | 257 | 14 (5.5) | 243 (94.5) | 27 (10.5) | 230 (89.5) |
| Divorced | 45 | 3 (6.7) | 42 (93.3) | 10 (22.2) | 35 (77.8) |
| Widowed | 8 | 2 (**25.0**) | 6 (75.0) | 3 (**37.5**) | 5 (62.5) |
| **Residence** | | | | | |
| Urban | 148 | 11 (7.4) | 137 (92.6) | 16 (10.8) | 132 (89.2) |
| Rural | 189 | 13 (6.9) | 176 (93.1) | 28 (14.8) | 161 (85.2) |

**Table 4. Prevalence of HPV (16/18) infections among participant women (n = 337) aged 21–49 years, with and without cervical lesions, Amhara region, Ethiopia, 2019.**

| Tissue type | Number (N = 337) | HPV positive | HPV negative |
|---|---|---|---|
| Cervical lesions | 44 | 17 (38.6%) | 27 (61.4%) |
| No cervical lesions | 293 | 7 (2.4%) | 286 (97.6%) |

women having primary education and below had heard about cervical cancer as compared to those women who had secondary school (84.9%) and college (94.12%).

Logistic regression analysis also revealed higher odds of cervical lesions among study participants who had HIV-positive status 17.9% vs 11.8%; COR = 1.62; 95% CI: 0.78–3.35) and had history of STIs (13.4% vs 12.3%; COR = 1.39; 95% CI: 0.64–3.00). In addition, slightly higher proportion of cervical lesions was also observed among study participants who resided in rural areas (18.8% vs 10.8%), and had $\geq$5 children (parity) (21% vs 8.7% of 3–4, 9.10% of 1–2 and 14.21 of null parity).

## Discussion

In this study we assessed HPV 16/18 infections, the presence of cervical lesions and associated factors among women aged 21 to 49 years in four hospitals in the Amhara region, Ethiopia. The overall prevalence of HPV and cervical lesions (precancerous and cancer lesions) in the study area was 7.1% and 13.1%, respectively. The observed HPV prevalence in the present study is lower compared to studies in another part of Ethiopia, by Leyh-Bannurah S. *et al.*, 17.3% [15], and by Ruland R. *et al.* in Attat hospital of rural Ethiopia, 16% [7]. This could mainly due to considerable differences in HPV detection methods between us and others. The antigen detection method we used detects only HPV 16 and 18 types, whereas others used molecular detection method for several HPV types. Also, the prevalence of presumed cervical lesion is lower than studies reported by Gedefaw A. *et al.* from southern Ethiopia 16.5% [16]. This is possibly due to the fact we excluded pregnant women and women with cervical cancers from our study. In support of this, studies show that HPV is the most common cause of viral cervical infection in pregnancy [17–20].

**Table 5. Logistic regression analysis of HPV positivity among participant women (n = 337) aged 21–49 years, Amhara region, Ethiopia, 2019.**

| Characteristic | Number (N = 337) | HPV infection based on E6 16/18 antigen test | | COR (95% CI) | p-value | AOR (95%CI) | p-value |
|---|---|---|---|---|---|---|---|
| | | Positive n (%) | Negative n (%) | | | | |
| **Age** | | | | | | | |
| 21–30 | 112 | 8 (7.1) | 104 (92.9) | 1 | | | |
| 31–40 | 158 | 9 (5.7) | 149 (94.3) | 0.78 (0.29–2.10) | 0.63 | | |
| 41–49 | 67 | 7 (**10.4**) | 60 (89.6) | 1.51 (0.52–4.39) | 0.44 | | |
| **Age of first intercourse** | | | | | | | |
| $\geq$18 | 106 | 4 (3.8) | 102 (96.2) | 1 | | | |
| <18 | 231 | 20 (**8.7**) | 211 (91.3) | 2.41 (0.80–7.25) | **0.11** | 1.31 (0.37–4.61) | **0.66** |
| **HIV** | | | | | | | |
| Negative | 270 | 22 (8.1) | 248 (91.9) | 1 | | | |
| Positive | 67 | 2 (3.0) | 65 (97.0) | 0.34 (0.07–1.51) | 0.15 | 0.31 (0.06–4.46) | 0.14 |
| **STI history** | | | | | | | |
| No | 276 | 18 (6.5) | 258 (93.5) | 1 | | | |
| Yes | 61 | 6 (**9.8**) | 55 (90.2) | 1.56 (0.59–4.11) | 0.36 | | |

**Table 6. Logistic regression analysis of VIA positivity among participant women (n = 337) aged 21–49 years, Amhara region, Ethiopia, 2019.**

| Characteristic | Number (N = 337) | VIA positivity | | COR (95% CI) | p-value | AOR (95%CI) | p-value |
|---|---|---|---|---|---|---|---|
| | | Positive n (%) | Negative n (%) | | | | |
| **Residence** | | | | | | | |
| Urban | 148 | 16 (10.8) | 132 (80.2) | 1 | | | |
| Rural | 189 | 28 (**14.8**) | 161 (85.2) | 1.43 (0.74–2.76) | 0.28 | | |
| **Education** | | | | | | | |
| Primary & below | 182 | 30 (16.5) | 152 (83.5) | 1 | | 1 | |
| Secondary | 53 | 8 (15.1) | 45 (84.9) | 0.90 (0.38–2.10) | 0.80 | 1.43 (0.11–18.17) | 0.78 |
| College & above | 102 | 6 (5.9) | 96 (94.1) | 0.31 (0.12–0.78) | **0.01** | 1.13 (0.01–110.24) | **0.95** |
| **Age of first intercourse** | | | | | | | |
| ≥18 | 106 | 8 (7.6) | 98 (92.4) | 1 | | | |
| <18 | 231 | 36 (**15.6**) | 195 (84.4) | 2.26 (1.0–5.05) | **0.04** | 1.88 (0.05–64.95) | **0.72** |
| **Parity** | | | | | | | |
| 0 | 35 | 5 (14.3) | 30 (85.7) | 1 | | | |
| 1–2 | 110 | 10 (9.1) | 100 (90.9) | 0.6 (0.19–1.89) | 0.38 | | |
| 3–4 | 92 | 8 (8.7) | 84 (91.3) | 0.57 (0.17–1.88) | 0.35 | | |
| ≥5 | 100 | 21(**21.0**) | 79 (79.0) | 1.59 (0.55–4.61) | 0.38 | | |
| **HIV status** | | | | | | | |
| Negative | 270 | 32 (11.9) | 238 (88.1) | 1 | | 1 | |
| Positive | 67 | 12 (**17.9**) | 55 (82.1) | 1.62 (0.78–3.35) | 0.19 | 5.47 (0.26–111.49) | **0.26** |
| **STI history** | | | | | | | |
| No | 276 | 34 (12.3) | 242 (87.7) | 1 | | | |
| Yes | 61 | 10 (**13.4**) | 51 (83.6) | 1.39 (0.64–3.00) | 0.39 | | |

Globally, HPV types 16 and 18 predominate and are responsible for most anogenital HPV-related cancers in women [21]. In our finding HPV prevalence among women without cervical lesions was 2.4% which is comparable with studies in Sudan (3.2%) [22], but much lower than other East Africa countries, Mozambique (40.3%) [23] and Kenya (41.4%) [24].

HPV prevalence among participant women with cervical lesions was 38.6%. Our finding is lower than from previous reports by Bekele A. *et al.* and Gemechu A. *et al.* in Ethiopia: Jimma (67.1%) and Hawasa (49.3%) [25, 26]. The lower prevalence of HPV in the present study can be justified as women of cervical cancer complaint visiting family planning or gynecology clinics were excluded from the present study. However, the proportion of VIA positivity in the present study is 13.1%, which is consistent with findings from different parts of Ethiopia, Addis Ababa, 13.1%, [27], Debremarkos, 14.1% [6] and Hawassa,12.9% [28].

The sensitivity and specificity of VIA in our study was 70.8% and 91.3% respectively which is low but comparable with studies by Arbyn M. et al. and Bobdey S. et al. [11, 12]. However, in our finding 7 out of 24 VIA negative women were identified positive by HPV OncoE6[TM] test which might indicate the importance of HPV testing together with VIA screening for early detection and follow up.

In accordance with the study by Abate S. *et al.* in Addis Ababa, Ethiopia [29], the prevalence of HPV infection and cervical lesions in the current study was similar across the different age groups with a peak in the age range of 40 to 49 years. These studies including ours suggest age is not an independent associated factor for HPV persistence infection, however, older women are more likely to develop persistent HR-HPV infections and cervical lesions [30].

Susceptibility to HPV infection is dependent on exposure but also on characteristics of the host such as immunosuppression and the presence of HIV. The present study, however, unlike

studies from Zambia [31] and Rwanda [32] revealed the low prevalence of HPV (3%) among HIV positive women than women without HIV (8%). This finding could be explained partly by exclusion of women with cervical cancer from the study. Also, the most common HR- HPV types among HIV-positive women would be non-HPV 16/18 types that could not be detected by the commercial HPV test kit used in the present study. On the contrary, HIV-positive participants had 1.62 times higher odds to develop cervical lesions than HIV negative, supporting HIV is an independent factor for cervical cancer [33]. Without antiretroviral therapy, HIV infection often leads to chronic immunosuppression, which in turn results in persistent HPV infection, a prerequisite to progression to cervical lesions. Nevertheless, given the small number of HIV-positive participants in the present study, future study on HIV-positive women is required to provide intervention strategies that would much benefit the HIV-positive women.

Unsafe sex is a primary transmission of genital human HPV, and STIs is an indicative of having unsafe sex. Concurrent to this, we found a slightly higher prevalence of HPV infections among women who had history of STIs (9.8%) than who had not (6.5%). The possible explanations might be because women co-infected with non-HPV STIs like chlamydia trachomatis, herpes simplex or genital warts could have altered epithelial tissue due to local inflammatory response which in turn make those women more susceptible to persistent HPV infection [34, 35].

Other cofactors are known to contribute for progression from cervical HPV infection to cervical cancer. In the present study, low level of education or being uneducated and high parity were found as associated factor for development of cervical lesions. Additionally, we found an association of cervical lesions with rural residence and history of STIs, albeit the association was not as such strong. In support of this, high proportion of uneducated and rural participants in this study never heard about cervical cancer, suggesting presence of health seeking behavior and knowledge gap in the region. A similar finding was reported from different studies in Ethiopia [36–38]. Women who had history of STIs were found 1.39 times higher odds to develop cervical lesions than women had no history of STIs. Similar finds were reported in studies from other regions of the Ethiopia [39].

Studies from Ethiopia, Africa and other parts of the world show that first sexual contact before age of 18 years is risk factor for cervical cancer [40–42]. Surprisingly, high proportion (69%) of participants in the present study reported having first sexual contact before the age 18 years, and of which 16% were diagnosed to have cervical lesions by VIA screening. This suggest early age sexual contact is also associated factor for cervical cancer, and the need to vaccinate female children at age 7–9 years before they have first sexual contact in an attempt to significantly reduce women deaths and mortality related to cervical cancer.

## Conclusions

Our study shows that prevalence of oncogenic HPV 16/18 and cervical lesions among women visiting hospitals for gynecology and family planning services are low. This is possibly because of (i) exclusion of pregnant women with confirmed cervical cancer and pregnancy from the study, and (ii) poor sensitivity of VIA. The present study indicates testing for HPV together with VIA improves early detection of high risk women for successful cervical cancer screening programs. However, more research is needed to better understand VIA positivity among HPV infected women. This study also identifies early age sexual contact related to early marriage, high parity and low level of education as independent associated factors to develop cervical lesions, supporting the importance of prioritizing the limited HPV screening test to those risk groups as well as vaccinating female children at age of 7–9 years.

## Limitation of the study

There are limitations in this study. First, the prevalence of cervical lesions in our study is more likely underestimated for the fact that high risk women were excluded from study, and possibly due to the low sensitivity of VIA to identify cervical lesions. Second, it is possible that our results regarding HPV infections are underestimated as the HPV E6 16/18 antigen detection test, employed in this study, detects only HPV 16 and 18 types and the low number of women enrolled in the study.

## Supporting information

**S1 Dataset. Dataset for prevalence of oncogenic human papillomavirus (HPV 16/18) infection, cervical lesions and its associated factors among women aged 21–49 years in Amhara region, Northern Ethiopia.**
(XLSX)

## Acknowledgments

We would like to thank study participants, staffs of study hospitals and Amhara Public Health Institute (APHI) staffs who had been providing constructive comments and technical support.

## Author Contributions

**Conceptualization:** Minwuyelet Maru Temesgen.

**Formal analysis:** Minwuyelet Maru Temesgen.

**Investigation:** Birtukan Shiferaw.

**Methodology:** Minwuyelet Maru Temesgen.

**Supervision:** Tefera Alemu, Birtukan Shiferaw, Seid Legesse.

**Writing – original draft:** Minwuyelet Maru Temesgen.

**Writing – review & editing:** Tefera Alemu, Seid Legesse, Taye Zeru, Mahteme Haile, Tesfaye Gelanew.

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
