## [Decision Letter · Decision Letter 0]

9 Nov 2020

PONE-D-20-26948

Prevalence of oncogenic human papilloma virus (HPV 16/18) infection, cervical cancer and its associated factors among women aged 21-49 years in Amhara region, Northern Ethiopia

PLOS ONE

Dear Dr. Temesgen,

Thank you for submitting your manuscript to PLOS ONE. After careful consideration, we feel that it has merit but does not fully meet PLOS ONE’s publication criteria as it currently stands. Therefore, we invite you to submit a revised version of the manuscript that addresses the points raised during the review process.

According to the expert reviews provided the manuscript would require additional revisions before being suitable for publication.

Please address the comments of individual reviewers with special emphasis on study participant selection, about the questionnaire administered, about the ethical questions and the description of the statistical analysis since all are a big part of the PLOS Publication criteria

https://journals.plos.org/plosone/s/criteria-for-publication

Furthermore, some additional issues to be addressed are listed below

We look forward to receiving your revised manuscript.

Kind regards,

Ivan Sabol

Academic Editor

PLOS ONE

Journal Requirements:

3. Thank you for including your ethics statement:

'The study was approved by APHI ethical review board ref. number 03/379/2011. All study subjects during the study period were informed about the purpose of the study and their consent was sought in written. Confidentiality of any information related to the participants was maintained using code numbers. Participants with a positive HPV test were linked to clinicians for further treatment and follow up.'

Additional Editor Comments:

1) It seems each VIA positive patient is considered as having cervical cancer. This doesn't appear to be accurate since VIA would be positive for pre-cancer lesions as well which the authors acknowledge at line 80. Please consider rephrasing the relevant text to include pre-cancer to VIA results for example at page 10 Line 168 and elsewhere

2) Line 128 which variables were included in the multivariable logistic regression?

3) Table 2 „Vaccinated for cc“ should be „Vaccinated for HPV“

4) Line 170 typo „HVP“ should be HPV

5) Line 174 and elsewhere please rephrase „abnormal cervical cytology“ to „cervical lesions“ or something similar since cytology was not performed

6) Tables 5 & 6 why were different variables considered for HPV or VIA positivity?

7) Table 6 the numbers for „Residence“ do not add up to 337

8) Sentence at line 223-224 is unclear. Possibly the authors meant „cervical cancer complaint“?

9) Language and grammar should be somewhat improved

Reviewers' comments:

Reviewer's Responses to Questions

**Comments to the Author**

1. Is the manuscript technically sound, and do the data support the conclusions?

Reviewer #1: Partly

Reviewer #2: Partly

2. Has the statistical analysis been performed appropriately and rigorously? 

Reviewer #1: Yes

Reviewer #2: Yes

3. Have the authors made all data underlying the findings in their manuscript fully available?

Reviewer #1: Yes

Reviewer #2: Yes

4. Is the manuscript presented in an intelligible fashion and written in standard English?

Reviewer #1: No

Reviewer #2: Yes

5. Review Comments to the Author

Reviewer #1: Reviewer’s comments

Full title: Prevalence of oncogenic human papillomavirus (HPV 16/18) infection, cervical cancer and its associated factors among women aged 21-49 years in Amhara region, Northern Ethiopia

Manuscript number: PONE-D-20-26948

Corresponding author: Minwuyelet Maru Temesgen Amhara Public Health Institute Dessie Branch Dessie, Amhara Region ETHIOPIA

Comments and questions

Overall comments

I would like to thank the authors for addressing potential factors for the emergence of health issues; cervical cancer among reproductive-age women in developing nations. Therefore, evidence regarding the prevalence of HPV is important to inform decision-making policy dealing with the prevention and control program of cervical cancers. The study presents the findings of original research in the area of women’s health and the results have not been published elsewhere. The study didn’t adequately perform relevant analyses with sufficient detail. The discussion part must comprehensively be written addressing policy, practical, methodological, and scientific implications. The article is presented not an acceptable level of English language standard and reporting guidelines.

Additional comments and questions

The author would respond to some of the following questions and comments.

1. Title: “Prevalence of oncogenic human papillomavirus (HPV 16/18) infection, cervical cancer and its associated factors among women aged 21-49 years in Amhara region, Northern Ethiopia”. The “…oncogenic human papillomavirus (HPV 16/18) infection…” is one of the risk factors for cervical cancer. But it's independently or separately mentioned in the topic. In the topic; there also “… it's associated factors…” though this is not clear to which condition these factors are considered to be associated. So, the author would restructure the title as…; “Prevalence of cervical cancer and its associated factors…”.

2. Abstract: Methods: Line 25: They would discuss sample calculation, sampling techniques, method of analysis used to address the prevalence and associated factors.

3. Abstract: conclusion: line 41 to 43: “…This study also identifies early age sexual contact, high parity and being uneducated/low educational status as primary risk factors to the development of cervical cancer…”. The study design of this study was “cross-sectional”, but the author tried to conclude the finding as “risk factors”. The risk factor is not a good expression or not appropriate for such a study design. Therefore; the author would replace phrases like “…associated factors”. See the topic.

4. Introduction: page: Line 75-78: “…In the Amhara region, testing for HPV infection has been not available or if available it is in limited areas, and hence little is known about the prevalence of HPV infection and cervical cancer, their co-prevalence’s and factors contributing to HPV infection and progression to cervical cancer”. The author mentioned the lack of HPV tests in the region as a justification to conduct the current study. This is not a good reason to conduct this study because the study has nothing to do with the availability or lack of HPV test facilities. Surprisingly; the study is conducted where the HPV test facilities are available. Can the author respond to this question?

5. Methods and materials: Study setting: Line 81: the author would briefly discuss the information the study settings that address health care system/health services in the region (health coverage, No.hospital, No. health center, coverage of cervical cancer care centers, HPV tests facilities, etc.)

6. Methods and materials: Study setting: Line 86-90: “All women who came for routine gynecologic or family planning services to those hospitals were used as a source population for this study. Women 88 age 21 to 49 years and referred to the cervical cancer screening services during the study period were included in the study. However, women with known cervical cancer and pregnancy were excluded”. This statement is talking about the study population and misplaced here. Therefore; the author would move it to the “study population section”.

7. Methods and materials: Sample size and sampling technique: Line 91: the authors should clarify how they draw/select the participants.

8. Methods and materials: data collection: Line 98: the interview technique must be clarified. Was it at the entry point or exit point?

9. Result part: How did the authors handle the relationship between HPV and cervical cancer?

10. Result: Table 5 (line 191), table 6 (line 201): “bivariate logistic regression…” what does it mean by “bivariate logistic regression”?. Since you are doing with multiple factors affecting the outcome variable?

11. Discussion: Line 213: “…The observed HPV prevalence in the present study is lower compared to studies in another part of Ethiopia, Gurage zone, 17.3%[12], and Atta hospital of rural Ethiopia, 16% [7]. Also, the prevalence of presumed cervical cancer is lower than studies reported from southern Ethiopia 16.5% [13]”. The authors tried to justify reasons as “…Knowledge about the prevalence of HPV among women with normal and abnormal cervical cytology is important to monitor and design HPV control programs…”. What does it mean? Did you explored the knowledge of women in this regard? If “yes”, was it high or low? Comment: the author would discuss the implication for health policy and practices. E.g the lower prevalence in your finding may be due to the inaccessibility of the services.

Reviewer #2: In general, I found this study interesting and relevant to the field. It is well designed and follow an appropriate methodology and analysis. However, there is unclarity on the consent procedure and participant selection to accept this manuscript as it is. I hope, the suggestions and questions below will help the authors to improve their manuscript.

Comments

Remove ‘this’ line 79

Q1, Please, clearly explain the inclusion and exclusion criteria. Line 87-90, “women age 21-49 years referred to the cervical cancer screening services during the study…”

Was this cervical screening service for the purpose of study or it was given as a routine service at all the four included hospitals?

Did the participant send it to the cervical cancer screening test after consent or before?

where was consent taken? Was at ANC, PMTCT, FP, or at the cancer screening center?

Q2, Ethical considerations, and consent procedures should be included in the body of the manuscript.

Q3, Participants also screened for HIV. What does screening mean? Were the participants tested for HIV? Or asked for self-reporting their HIV status? If tested, did they counseled? who performed the testing? How did you manage refusals?

Q4, why did you use a design effect in this study? The design effect is used for multistage sampling studies. But this study is a facility-based cross-sectional study.

Q5, Modify table name for table 1. – “Demographic characteristics of study participants, women aged 21-49 years…”,. The Demographic characteristics are not about the whole Amahar region. Follow an appropriate table naming and apply this for all other tables.

Q6, Remove the total raw end of table 1. It is confusing and it is better to put n=337 at the headers of the table as Frequency(N= 337). Apply this for all other tables.

Q7, Line 151, replace “had not heard” by never heard about

Q8, Line 224: “visiting family planning or gynecology clinics 224 were excluded from the present study” Was this included in the exclusion criteria? And justify the reason why the women were excluded.

6. PLOS authors have the option to publish the peer review history of their article (what does this mean?). If published, this will include your full peer review and any attached files.

Reviewer #1: **Yes: **Fira Abamecha

Reviewer #2: **Yes: **Serebe Gebrie

---

## [Author Response · Author response to Decision Letter 0]

13 Dec 2020

Response to the Editor and Reviewers:

Dear Editor and Reviewers!

The authors would like to thank the area editor and the reviewers for their precious time and invaluable comments to our submitted article “Prevalence of oncogenic human papilloma virus (HPV 16/18) infection, cervical cancer and its associated factors among women aged 21-49 years in Amhara region, Northern Ethiopia”. We greatly appreciate the thorough and thoughtful comments provided to our paper. We are resubmitting the revised version according to the comments and we believe that the revised version of our paper addresses all concerns. Minor revisions and grammatical corrections are applied to the manuscript modification along with revisions explained in this document. Attached below are detailed responses to the comments. 

Response to the Editor

1) It seems each VIA positive patient is considered as having cervical cancer. This doesn't appear to be accurate since VIA would be positive for pre-cancer lesions as well which the authors acknowledge at line 80. Please consider rephrasing the relevant text to include pre-cancer to VIA results for example at page 10 Line 168 and elsewhere

Response: It is addressed

• VIA can be positive in precancerous or cancerous lesions and therefore pre-cancer lesion is added to relevant section in the revised version.

2) Line 128 which variables were included in the multivariable logistic regression?

Response: We appreciate the query. Variables with p-value less than 0.2 in binary logistic regression were included in multivariable logistic regression. Based on this we have now mentioned as indicated in line 138 of the revised manuscript

• For association of HPV acquisition; age of first sexual intercourse, HIV status, occupation and marital status were included in multivariable logistic regression 

• For association of VIA positivity; age, age of first sexual intercourse, HIV status, educational status, occupation, time of oral contraceptive use and marital status were included in multivariable logistic regression

3) Table 2 „Vaccinated for cc“ should be „Vaccinated for HPV“

Response: Corrected

4) Line 170 typo „HVP“ should be HPV

Response: Corrected

5) Line 174 and elsewhere please rephrase „abnormal cervical cytology“ to „cervical lesions“ or something similar since cytology was not performed

Response: Thank you for this important comment. We rephrase abnormal cervical cytology to cervical lesions 

6) Tables 5 & 6 why were different variables considered for HPV or VIA positivity?

Response: The variables are selected based on the assumption and certain evidences that factors for acquiring HPV infection are different from factors for developing cervical cancer. Therefore, variables in table 5 (current age, age of first sexual intercourse, HIV status and history of sexually transmitted illness) are related to sexual behavior which could affect acquisition of HPV, whereas, variables in table 6 (age of first sexual intercourse, HIV status, residence, history of STI, educational status and parity) were considered as factors for cervical cancer.

7) Table 6 the numbers for „Residence“ do not add up to 337

Response: It is addressed

8) Sentence at line 223-224 is unclear. Possibly the authors meant „cervical cancer complaint“?

Response: It is addressed

9) Language and grammar should be somewhat improved

• It is done

Response to Reviewer #1 

1. Title: “Prevalence of oncogenic human papillomavirus (HPV 16/18) infection, cervical cancer and its associated factors among women aged 21-49 years in Amhara region, Northern Ethiopia”. The “…oncogenic human papillomavirus (HPV 16/18) infection…” is one of the risk factors for cervical cancer. But it's independently or separately mentioned in the topic. In the topic; there also “… it's associated factors…” though this is not clear to which condition these factors are considered to be associated. So, the author would restructure the title as…; “Prevalence of cervical cancer and its associated factors…”.

Response: Thank you for this suggestion. As the reviewer points out, oncogenic human papilloma virus infection is one of the risk factor for cervical cancer; however our study was also aimed to describe the prevalence of HPV with associated factors for infection. By considering this, we believe the title as it is would be more explanatory.

2. Abstract: Methods: Line 25: They would discuss sample calculation, sampling techniques, method of analysis used to address the prevalence and associated factors.

Response: Thank you for this comment. It is addressed

• The sample calculation used was single population proportion formula and women were consecutively recruited (page 2, line 29, line111). Sixteen percent HPV prevalence, 5% margin of error and 1.5 design effect was considered for the calculation. (Line 111)

3. Abstract: conclusion: line 41 to 43: “…This study also identifies early age sexual contact, high parity and being uneducated/low educational status as primary risk factors to the development of cervical cancer…”. The study design of this study was “cross-sectional”, but the author tried to conclude the finding as “risk factors”. The risk factor is not a good expression or not appropriate for such a study design. Therefore; the author would replace phrases like “…associated factors”. See the topic.

Response: It is addressed. 

• Risk factor is replaced with associated factor

4. Introduction: page: Line 75-78: “…In the Amhara region, testing for HPV infection has been not available or if available it is in limited areas, and hence little is known about the prevalence of HPV infection and cervical cancer, their co-prevalence’s and factors contributing to HPV infection and progression to cervical cancer”. The author mentioned the lack of HPV tests in the region as a justification to conduct the current study. This is not a good reason to conduct this study because the study has nothing to do with the availability or lack of HPV test facilities. Surprisingly; the study is conducted where the HPV test facilities are available. Can the author respond to this question?

Response: We sincerely appreciate the comment. The reviewer mention lack of HPV test or availability in limited area cannot be justification to conduct the current study. 

• We would like to mean due in many health facilities of the region, the prevalence remain unknown (line 84) and also indicated in the method. The current study provided HPV test kit for the study purpose to selected hospitals which have HPV testing setups (testing machine). Although the study has nothing to do (or plan to do) with the availability of HPV tests in health facilities, by revealing the facts about the prevalence and associated factors, it recommend to scale up HPV testing to every facility in the region.

5. Methods and materials: Study setting: Line 81: the author would briefly discuss the information the study settings that address health care system/health services in the region (health coverage, No.hospital, No. health center, coverage of cervical cancer care centers, HPV tests facilities, etc.)

Response: We acknowledge this information pointed out by the reviewer

• We added description of the study setting, the health care system and cervical cancer services in the region to the study setting ( Line 95)

6. Methods and materials: Study setting: Line 86-90: “All women who came for routine gynecologic or family planning services to those hospitals were used as a source population for this study. Women 88 age 21 to 49 years and referred to the cervical cancer screening services during the study period were included in the study. However, women with known cervical cancer and pregnancy were excluded”. This statement is talking about the study population and misplaced here. Therefore; the author would move it to the “study population section”.

Response: It is addressed and moved to study population (Line 101

7. Methods and materials: Sample size and sampling technique: Line 91: the authors should clarify how they draw/select the participants.

Response: It is addressed.

• The sample size is calculated as indicated in comment 2 and study participants were recruited consecutively 

8. Methods and materials: data collection: Line 98: the interview technique must be clarified. Was it at the entry point or exit point?

Response: We apologize for our lack of clarity. It is clarified in the revised version (Line 115).

• All eligible clients were screened for cervical lesions using VIA and tested for HPV. Study participants were recruited at the exit point for interview and review of medical records to get VIA and HPV results after getting informed consent.

9. Result part: How did the authors handle the relationship between HPV and cervical cancer?

Response: It is an established fact that HPV infection is a risk factor for cervical cancer. In our study we focused on describing the prevalence of oncogenic HPV and cervical cancer or precancerous lesions separately with possible associated factors for HPV acquisition and cervical lesions, thus HPV and cervical cancer association was not described.

10. Result: Table 5 (line 191), table 6 (line 201): “bivariate logistic regression…” what does it mean by “bivariate logistic regression”?. Since you are doing with multiple factors affecting the outcome variable?

Response: We thank the reviewer for pointing out the type of analysis in relation to factors affecting the outcome variable. We have considered the comment and both crude odds ratio (COR) and adjusted odds ratio (AOR) was used to describe the association in table 5 (Line 215) and table 6 (Line 246)

11. Discussion: Line 213: “…The observed HPV prevalence in the present study is lower compared to studies in another part of Ethiopia, Gurage zone, 17.3%[12], and Atta hospital of rural Ethiopia, 16% [7]. Also, the prevalence of presumed cervical cancer is lower than studies reported from southern Ethiopia 16.5% [13]”. The authors tried to justify reasons as “…Knowledge about the prevalence of HPV among women with normal and abnormal cervical cytology is important to monitor and design HPV control programs…”. What does it mean? Did you explored the knowledge of women in this regard? If “yes”, was it high or low? Comment: the author would discuss the implication for health policy and practices. E.g the lower prevalence in your finding may be due to the inaccessibility of the services.

Response: The sentence “…Knowledge about the prevalence of HPV….” was intended for description of knowledge about HPV among women with and without cervical lesion and it is another concept not put to justify the low prevalence of HPV, thus was deleted for its ambiguity and the reason for low prevalence is justified; variation in detection methods ( in line 260), exclusion of pregnant women and cervical cancer cases (264). 

Response to Reviewer #2 

Comments

Remove ‘this’ line 79

Response:- It is done

Q1, Please, clearly explain the inclusion and exclusion criteria. Line 87-90, “women age 21-49 years referred to the cervical cancer screening services during the study…”

Was this cervical screening service for the purpose of study or it was given as a routine service at all the four included hospitals?

Did the participant send it to the cervical cancer screening test after consent or before?

where was consent taken? Was at ANC, PMTCT, FP, or at the cancer screening center?

Response: Thank you for pointing this out. 

• Cervical screening service was given as a routine service in all hospitals. But for the purpose of the study HPV test kit was provided and all eligible clients screened for cervical lesions were also sent to test for HPV. Study participants were recruited at the exit point, after getting the service (VIA and HPV test). Consent was sought for interview and review of medical records to get VIA and HPV and HIV results. (Data collection: Line 115). 

• The inclusion criteria: Women of age 21 to 49 years eligible for cervical cancer screening visiting the clinics for routine services during the study period were included (Line 104)

• Exclusion criteria: women with known cervical cancer and pregnancy were excluded (line 106).

Q2, Ethical considerations, and consent procedures should be included in the body of the manuscript.

Response: It is done (Line 333)

Q3, Participants also screened for HIV. What does screening mean? Were the participants tested for HIV? Or asked for self-reporting their HIV status? If tested, did they counseled? who performed the testing? How did you manage refusals?

Response: HIV testing is part of cervical cancer screening program and the result is obtained from medical records. The ethical issue was addressed in the consent.

Q4, why did you use a design effect in this study? The design effect is used for multistage sampling studies. But this study is a facility-based cross-sectional study.

Response: Thank you for this comment. Since the hospitals are found in different zonal administrations of the region, they are considered as clusters. Therefore, even though the study was cross sectional, design effect was used.

Q5, Modify table name for table 1. – “Demographic characteristics of study participants, women aged 21-49 years…”,. The Demographic characteristics are not about the whole Amahar region. Follow an appropriate table naming and apply this for all other tables.

Response: It is done

Q6, Remove the total raw end of table 1. It is confusing and it is better to put n=337 at the headers of the table as Frequency(N= 337). Apply this for all other tables.

Response: it is done

Q7, Line 151, replace “had not heard” by never heard about

Response: It is done

Q8, Line 224: “visiting family planning or gynecology clinics 224 were excluded from the present study” Was this included in the exclusion criteria? And justify the reason why the women were excluded.

Response: We apologize for our lack of clarity. The exclude ones were women of cervical compliant visiting the clinics. It is corrected

---

## [Decision Letter · Decision Letter 1]

9 Feb 2021

PONE-D-20-26948R1

Prevalence of oncogenic human papillomavirus (HPV 16/18) infection, cervical lesions and its associated factors among women aged 21-49 years in Amhara region, Northern Ethiopia

PLOS ONE

Dear Dr. Temesgen,

Thank you for submitting your manuscript to PLOS ONE. After careful consideration, we feel that it has merit but does not fully meet PLOS ONE’s publication criteria as it currently stands. Therefore, we invite you to submit a revised version of the manuscript that addresses the points raised during the review process.

While most of the original comments were addressed, some additional issues were noted that should be improved or at least the limitations highlighted.

We look forward to receiving your revised manuscript.

Kind regards,

Ivan Sabol

Academic Editor

PLOS ONE

Reviewers' comments:

Reviewer's Responses to Questions

**Comments to the Author**

1. If the authors have adequately addressed your comments raised in a previous round of review and you feel that this manuscript is now acceptable for publication, you may indicate that here to bypass the “Comments to the Author” section, enter your conflict of interest statement in the “Confidential to Editor” section, and submit your "Accept" recommendation.

Reviewer #2: All comments have been addressed

Reviewer #3: All comments have been addressed

Reviewer #4: All comments have been addressed

2. Is the manuscript technically sound, and do the data support the conclusions?

Reviewer #2: Yes

Reviewer #3: Partly

Reviewer #4: Partly

3. Has the statistical analysis been performed appropriately and rigorously? 

Reviewer #2: Yes

Reviewer #3: I Don't Know

Reviewer #4: Yes

4. Have the authors made all data underlying the findings in their manuscript fully available?

Reviewer #2: Yes

Reviewer #3: Yes

Reviewer #4: Yes

5. Is the manuscript presented in an intelligible fashion and written in standard English?

Reviewer #2: No

Reviewer #3: Yes

Reviewer #4: Yes

6. Review Comments to the Author

Reviewer #2: Revised manuscript review

The authors have addressed all my concerns point-by-point and made a considerable revision. The manuscript is now greatly improved, most of the suggestions are included, and other issues are satisfactorily justified. However, minor editorial corrections are still needed but can be considered for publication after proofreading and minor editorial correction have done without further review of the revised version.

For example,

Abstract page 2, L29-31, there is a repetition of the whole sentence.

L94 – avoid spacing before a comma,

L73-74 – Revise the sentence ending with "in all most all cervical cancer cases".

L351-449 – Check reference writing(the font and spacing) as per the journal requirement. Unlike the body of the paper, the reference seems written in smaller font and without spacing.

Reviewer #3: Major concern concerning this study

In the Introduction section, the authors should elaborate in what consist the cervical cancer prevention programs in Ethiopia and provide relevant references. What is the status of cervical screening, VIA, HPV testing, and HPV vaccination? What is the particularity of Amhara region of Northern Ethiopia compared to the rest of the country? This study did not bring any particular knowledge that is already established in many other countries. So, the authors should focus on a particular goal and emphasize what novelty this study brings to the scientific community. In addition, the major limitation of the study is the low number of enrolled women (N=337), therefore the conclusion is unreadable. The authors should enlarge they study to at least few thousands respondents.

Minor, but essential, correction to do

Abstract, Methods, line 23-27. Unnecessary repetition.

Introduction, line 64: “and late detection of HPV infection” should be “and late detection of cervical lesion”

Introduction, line 86: What does it stand for “cervical cancer prevention programs”?

Introduction, line 79: “where HPV testing is unavailable” should be “where cervical cancer screening by cervical cytology and HPV testing is unavailable”

Results: all numbers should be with one decimal.

Discussion: when referring to others studies be precise, i.e. Author et al. When comparing your method of testing to others be precise as well and cite the appropriate reference.

Reviewer #4: The focus of the study article is very important to Ethiopia and other developing countries still using VIA as a cervical cancer screening strategy. However, the uptake of VIA is low and there are a lot of issues associated with its sensitivity and specificity. Besides, quality control is another grey area in the implementation of VIA as a primary screening strategy. Onco6 HPV antigen test is one of the HPV tests available in the market with high specificity for lesions associated with HPV 16 and 18 but its sensitivity is low as a test and this is the inherent problem of the test. As close to 70% of cervical cancer is associated with HPV16 and 18 the test is assumed to be one of the good options in the provision of HPV based cervical cancer screening service. However, as the purpose of cervical cancer screening is to identify women with pre-cancerous lesion and viruses other than HPV 16 and 18 maybe important in the study settings thus this particular study will miss significant number of high-risk HPV infections. Thus, with the test type employed the authors will be able to identify only persistent HPV16 and 18 infection and not the prevalence of high-risk HPV and this needs to be corrected in their text. Although VIA is advocated for its specificity by those who recommend the test for primary screening, VIA missed seven out of 24 Onco6 test positives and 27 of VIA positives were Onco6 test negatives. Of course, about seven of the 27 Onco6 test negatives had STI history. So many factors account for the progression of the HPV infection to cervical cancer, HPV genotype is one and they were only able to identify HPV 16 and 18 and not others by the method they have employed. As cervical cancer is a kind of progressive disease identifying factors associated with progression of HPV infection cannot be met by such correctional study design. In summary, the finding of this study is publishable if the authors rather show how VIA is neither a sensitive nor a specific test to identify pre-cancerous lesions. The authors need further to look at their data justify why 61% (27/44) of the VIA positive were Onco6 negative and recommend further study to assess the utility of VIA to identify specific pre-cancerous lesions among those who are Onco6 test positives.

7. PLOS authors have the option to publish the peer review history of their article (what does this mean?). If published, this will include your full peer review and any attached files.

Reviewer #2: **Yes: **Serebe Gebrie

Reviewer #3: No

Reviewer #4: **Yes: **Tamrat Abebe

---

## [Author Response · Author response to Decision Letter 1]

4 Mar 2021

Response to the Reviewers:

Dear Reviewers!

The authors would like to thank the reviewers for their precious time and vital comments to our submitted article titled “Prevalence of oncogenic human papilloma virus (HPV 16/18) infection, cervical lesions and its associated factors among women aged 21-49 years in Amhara region, Northern Ethiopia” (PONE-D-20-26948R1). We greatly appreciate the comments and we are resubmitting the revised version. We believe the revised version addresses all concerns and attached below are detailed responses to the comments. 

Response to Reviewer #2 

1. Abstract page 2, Line 29-31, there is a repetition of the whole sentence

Response: - sorry for the repetition, It is corrected

2. Line 94: Avoid spacing before comma

Response: Thank you for this comment. It is addressed

3. Line 73-74 Revise the sentence ending with “in all most all cervical cancer cases”.

Response: It is addressed and corrected as “in almost all cervical cancer cases”

4. Line 351-499: Reference writing (font and spacing)

Response:- Thank you, the font and spacing is corrected

Response to Reviewer #3 

• Cervical cancer prevention programs in Ethiopia and the status of VIA, HPV testing and vaccination in Ethiopia and Amhara region should be elaborated. In addition, low number of enrolled women is major limitation

Response: Thank you for this point. 

The content of cervical cancer prevention and control program in Ethiopia is included in the introduction based on 2015 Ethiopian Ministry of health cervical cancer prevention and control guideline in Line 80, Line 87-96 of the revised manuscript and the low sample size is written in the limitation of the study.

• Abstract Method line 23-27, unnecessary repetition

Response: It is addressed 

• Introduction line 64 “and late detection of HPV infection” should be “and late detection of cervical lesion”

Response: It is corrected 

• Introduction line 86 what does it stand for “cervical cancer prevention program”

Response: Thank you for this comment. It is elaborated based on the local context.

• Introduction line 79: “where HPV testing is unavailable” should be “where cervical cancer screening by cervical cytology and HPV testing is unavailable” 

Response: It is done

• Result: all numbers should be in one decimal

Response: it is done

• Discussion: when referring to others studies be precise, Author et al. When comparing your method of testing to others be precise as well and cite the appropriate reference 

Response: It is addressed

Response to Reviewer #4

The focus of the study article is very important to Ethiopia and other developing countries still using VIA as a cervical cancer screening strategy. However, the uptake of VIA is low and there are a lot of issues associated with its sensitivity and specificity. Besides, quality control is another grey area in the implementation of VIA as a primary screening strategy. Onco6 HPV antigen test is one of the HPV tests available in the market with high specificity for lesions associated with HPV 16 and 18 but its sensitivity is low as a test and this is the inherent problem of the test. As close to 70% of cervical cancer is associated with HPV16 and 18 the test is assumed to be one of the good options in the provision of HPV based cervical cancer screening service. However, as the purpose of cervical cancer screening is to identify women with pre-cancerous lesion and viruses other than HPV 16 and 18 maybe important in the study settings thus this particular study will miss significant number of high-risk HPV infections. Thus, with the test type employed the authors will be able to identify only persistent HPV16 and 18 infection and not the prevalence of high-risk HPV and this needs to be corrected in their text. Although VIA is advocated for its specificity by those who recommend the test for primary screening, VIA missed seven out of 24 Onco6 test positives and 27 of VIA positives were Onco6 test negatives. Of course, about seven of the 27 Onco6 test negatives had STI history. So many factors account for the progression of the HPV infection to cervical cancer, HPV genotype is one and they were only able to identify HPV 16 and 18 and not others by the method they have employed. As cervical cancer is a kind of progressive disease identifying factors associated with progression of HPV infection cannot be met by such correctional study design. In summary, the finding of this study is publishable if the authors rather show how VIA is neither a sensitive nor a specific test to identify pre-cancerous lesions. The authors need further to look at their data justify why 61% (27/44) of the VIA positive were Onco6 negative and recommend further study to assess the utility of VIA to identify specific pre-cancerous lesions among those who are Onco6 test positives.

Response: Thank you for pointing out issues related to 

- persistent HPV16 and 18 infection versus high-risk HPV

- the uptake, sensitivity and quality control of VIA,

- the importance of HPV genotypes other than 16 and 18, 

- persistent HPV 16 and HPV 18 infection versus prevalence of high-risk HPV 

The sensitivity and specificity of VIA was calculated and added to the result section (line 213-216). VIA identified 17 and missed 7 out of 24 HPV positive women that make its sensitivity 70.8% and 286 out of 293 participant women were correctly identified negative which make VIA specificity 91.3%. However, the positive predictive value of VIA is only 61.4% (27 out of 44). The low sensitivity of VIA is indicated in the limitation and further study is recommended (Line 348)

Regarding the HPV genotypes our study aimed to HPV 16 and HPV 18 which are considered highly prevalent genotypes associated with cervical cancer. Since women visiting gynecology and family planning service were recruited for the study, it is believed to be prevalence.

---

## [Editor Report · Decision Letter 2]

9 Mar 2021

Prevalence of oncogenic human papillomavirus (HPV 16/18) infection, cervical lesions and its associated factors among women aged 21-49 years in Amhara region, Northern Ethiopia

PONE-D-20-26948R2

Dear Dr. Temesgen,

We’re pleased to inform you that your manuscript has been judged scientifically suitable for publication and will be formally accepted for publication once it meets all outstanding technical requirements.

Kind regards,

Ivan Sabol

Academic Editor

PLOS ONE

Additional Editor Comments (optional):

During revision some typos were introduced or missed. Please address these in particular and/or any other language or typographical issues.

Line 84 „prior to sexual debut using was started“ - missing words

Line 142 „OncoE6TM“ TM should probably be in superscript throughout the manuscript. For the test the manufacturer "Arbor Vita Corporation, Fremont, CA, USA" should be listed, not a commercial vendor.

Line 93 "Nonthless," typo

Line 143 „manufacture’s“ typo

Line 237 "who were educated above college had" - grammatically incorrect replace with "with college degree or higher education had"

Line 238 "educated primary and below." - grammatically incorrect replace with "women who had primary education or less".

Line 268 "13.4% vs12.3%;" no space between vs and the number

Line 299 „onco6 test“ is should be stated as before as „OncoE6 test“
---

## [Editor Report · Acceptance letter]

11 Mar 2021

PONE-D-20-26948R2 

Prevalence of oncogenic human papillomavirus (HPV 16/18) infection, cervical lesions and its associated factors among women aged 21-49 years in Amhara region, Northern Ethiopia. 

Dear Dr. Temesgen:

I'm pleased to inform you that your manuscript has been deemed suitable for publication in PLOS ONE. Congratulations! Your manuscript is now with our production department. 

Kind regards, 

on behalf of

Dr. Ivan Sabol 

Academic Editor

PLOS ONE